# The uptake of WHO-recommended birth preparedness and complication readiness messages during pregnancy and its determinants among Ethiopian women: A multilevel mixed-effect analyses of 2016 demographic health survey

**Aklilu Habte** ID *, **Aiggan Tamene** ID, **Demelash Woldeyohannes**

School of Public Health, College of Medicine and Health Sciences, Wachemo University, Hosanna, Ethiopia

* akliluhabte57@gmail.com

**Data Availability Statement:** The raw dataset used and analyzed in this study can be accessed from

## Abstract

### Background

Birth preparedness and complication readiness (BPCR) is a package of interventions recommended by the World Health Organization to improve maternal and newborn health and it is provided and implemented through a focused antenatal care program. This study aimed at assessing the uptake of birth preparedness and complication readiness messages, and compliance with each key message, among Ethiopian women during their recent pregnancies using the 2016 demographic health survey report.

### Methods

The data for this study was taken from the Ethiopian Demographic and Health Survey, which was conducted from January to June 2016 and covered all administrative regions. STATA version 16 was used to analyze a total of 4,712 (with a weighted frequency of 4,771.49) women. A multilevel mixed-effects logistic, and multilevel mixed-effect negative binomial regressions were fitted, respectively. Adjusted odds ratio (AOR) and Incidence rate ratio (IRR) with their corresponding 95% confidence interval (CI) were used to report significant determinants.

### Results

More than half, 56.02% [95% CI: 54.58, 57.41] of women received at least one birth preparedness and complication readiness message. Being in the richest wealth quintiles (AOR = 2.33; 95% CI: 1.43, 3.73), having two birth/s in the last five years (AOR = 1.54; 95% CI: 1.13, 2.10), receiving four or more antenatal visits(AOR = 3.33; 95% CI: 2.49, 4.45), and reading a newspaper at least once a week (AOR = 1.27; 95% CI: 1.07, 1.65) were the individual-level factors, whereas regions and residence(AOR = 1.54; 95% CI: 1.11, 1.96) were

the DHS website (http://www.measuredhs.com) based on a reasonable request.

**Funding:** The author(s) received no specific funding for this work.

**Competing interests:** The authors have declared that no competing interests exist.

**Abbreviations:** AIC, Akaike's information criterion; ANOVA, analysis of variance; ANC, Antenatal Care; AOR, Adjusted odds ratio; BPCR, Birth preparedness and complication readiness; COR, Crude odds ratio; CSA, Central Statistical Agency; DHS, Demographic and health survey; EA, Enumeration area; EDHS, Ethiopian Demographic and Health Survey; ICC, Intra Class Correlation Coefficient; IRR, Incidence Rate Ratio; NDSs: Neonatal Danger Signs; PCV, Proportional Change in Variance; PNC, Postnatal Care; PPP, Postpartum Period; SDG, Sustainable Development Goal; WHO, World Health Organization.

the community-level factors associated with the uptake of at least one BPCR message. On the other hand, receiving four or more antenatal visits (IRR = 2.78; 95% CI: 2.09, 3.71), getting permission to go to a health facility (IRR = 1.29; 95% CI: 1.028, 1.38), and not covered by health insurance schemes (IRR = 0.76; 95% CI: 0.68, 0.95) were identified as significant predictors of receiving key birth preparedness and complication readiness messages.

## Conclusion

The overall uptake of the WHO-recommended birth readiness and complication readiness message and compliance with each message in Ethiopia was found to be low. Managers and healthcare providers in the health sector must work to increase the number of antenatal visits. Policymakers should prioritize the implementation of activities and interventions that increase women's autonomy in decision-making, job opportunity, and economic capability to enhance their health-seeking behavior. The local administrative bodies should also work to enhance household enrollment in health insurance schemes.

## Introduction

Maternal mortality is a significant problem in developing countries, and reducing maternal mortality has gained global recognition, as evidenced by its inclusion in the Sustainable Development Goals (SDG) [1]. Every pregnancy carries risks, and a woman dies every minute of every day as a result of complications during pregnancy and childbirth. The majority of these deaths could be avoided if women had access to quality maternal health care [2]. Sub-Saharan Africa accounted for nearly two-thirds of maternal deaths, accounting for 99% of maternal deaths worldwide [3]. Ethiopia, a Sub-Saharan African country, has a high maternal mortality rate of 412 maternal deaths per 100,000 live births, as well as low coverage of ANC, skilled delivery, and immediate postpartum care at 62%, 26%, and 13%, respectively [4].

Preventable direct obstetric causes accounted for more than three-quarters of the global maternal mortality burden [5]. Globally, five major complications tally for more than 70% of all maternal deaths: hemorrhage, infection, unsafe abortion, hypertensive disorders of pregnancy, and obstructed labor [1, 6]. During or following pregnancy and childbirth, an estimated 40% of pregnant women (50 million per year) experienced pregnancy-related health problems, with 15% experiencing serious or long-term complications [7]. As a result, 300 million women experience pregnancy-related health complications and disabilities such as anemia, uterine prolapse, fistula, pelvic inflammatory disease, and infertility [8, 9].

Many of the complications that result in maternal and perinatal deaths are unpredictable, and their onset can be both sudden and severe [10]. Safe motherhood initiatives focusing on antenatal care, clean delivery, and postnatal care, as well as Making Pregnancy Safer, were lists of strategies implemented to reduce those tragic maternal deaths [11]. Birth preparedness and complication readiness (BPCR) is one of the WHO recommendations on maternal and newborn health promotion interventions that are provided and implemented through a focused antenatal care program [11, 12]. It is a key component of the safe motherhood strategy, intending to encourage pregnant women and their families to use skilled maternal and neonatal care during childbirth by designing a birth plan and encouraging active preparation and decision-making for delivery [13, 14]. It encourages women, households, and communities to make arrangements such as identifying or establishing available transport, setting aside money to

pay for service fees and transport, and identifying blood donors to facilitate swift decision-making and reduce delays in reaching care once a problem arises [12, 15].

Too often, however, their access to care is impeded by delays; delays in deciding to seek care, delays in reaching care, and delays in receiving care. These delays have many causes; including logistic and financial concerns, unsupportive policies, and gaps in services, as well as inadequate community and family awareness and knowledge about obstetric complication issues [16–21].

Unlike other maternal health services, little is known about BPCR practice in Ethiopia. Although BPCR is essential for further improving maternal and child health and preventing maternal deaths, the Ethiopian government and healthcare providers have placed little emphasis on it. To the best of our knowledge, no study report exists that describes the magnitude and determinants of compliance with WHO-recommended BPCR messages at the national level. Hence, the purpose of this study was to assess the level of BPCR message uptake in Ethiopia using the EDHS 2016 report. This study will help to provide evidence-based knowledge on the level and determinants of BPCR, as well as basic data for service providers, policymakers, and programmers to design effective BPCR program interventions to reduce high maternal and neonatal mortality rates.

## Methods and materials

### Study setting, data, and period

The study relied on population-based, nationally representative data from the 2016 Ethiopian Demographic and Health Survey (DHS), the fourth in a series of standardized national-level population and health surveys carried out as part of the global Demographic and Health Survey (DHS) program [4]. Ethiopia is in North-eastern (horn of) Africa, between 3˚ and 15˚ North latitude and 33˚ 48˚ and East longitudes. Ethiopia's healthcare system is divided into three levels: primary, secondary, and tertiary care. Primary hospitals, health centers, and health posts provide primary care, general hospitals provide secondary care, and specialized hospitals provide tertiary care. The survey was conducted from January 18, 2016, to June 27, 2016, by the Central Statistical Agency (CSA) in collaboration with the Federal Ministry of Health (FMOH) and the Ethiopian Public Health Institute, with technical assistance from ICF International and financial support from USAID, the government of the Netherlands, the World Bank, Irish Aid, and UNFPA. Data of the study participants were accessed on October 23, 2022 from DHS website, their URL: www.dhsprogram.com by contacting them via personal email communication with a possible justification for the data request. Permission was granted via email after reviewing the account. A cross-sectional study design using secondary data from 2016 EDHS was conducted.

### The population of the study

The source population consisted of 15,683 women who had given birth within five years preceding the survey. The study population consisted of 4,712 women who had complete information on the uptake of BPCR messages during their ANC visit, as well as the contents of those messages, and the entire analyses were conducted on them. Due to a lack of information on service uptake, a total of 10,971 respondents were excluded from the analysis (Missing values).

### Sampling procedures

The 2007 Ethiopia Population and Housing Census sampling frame was used, which included 84,915 enumeration areas (EAs), with each EA covering 181 households. A stratified two-stage

cluster design was used to select respondents, as each region was stratified into urban and rural areas. The first step was to select 645 clusters (202 urban and 443 rural areas) with a probability proportional to the size of the enumeration area and independent selection within each stratum. The household listing was completed in all of the selected EAs between September and December 2015. The second stage involved the selection of 28 households per cluster using an equal probability systematic selection of eligible women aged 15–49 years. With a response rate of 94.6%, a sample of 16,650 households and 15,683 women aged 15–49 years was identified. Furthermore, the survey design and methodology were detailed in the 2016 EDHS [4].

## Measurement of variables of the study

**Outcome variables.** This study had two outcome variables.

*The first outcome variable.* Was the receipt of BPCR messages during ANC visits which was assessed by the question "During any of your antenatal visits were you told about BPCR?" If a mother said "Yes," the response was labeled as 1, otherwise it was labeled as 0.

*The second outcome variable.* Was the number of WHO-recommended BPCR messages received by a mother during pregnancy which was assessed using six items. Those key messages were about determining the place of birth, obtaining necessary supplies for childbirth, preparing emergency transportation, saving money for emergency expenses, identifying companions during labor and childbirth, and securing potential blood donors. Information on these six key BPCR messages was derived from the response to the question: Were you told about your Place of birth? Were you told about supplies needed for birth? Were you told about emergency transportation?. . . The answers were recorded as Yes (= 1) or No (= 0). During the same pregnancy, a single mother may be informed about the place of birth or the supplies required for childbirth several times. However, because the mother was asked to report any messages she received at least once, any response was recorded as a single message. Based on the responses, a composite index of BPCR was created, which is simply a count of the number of key messages received. The variable had a minimum value of zero indicating that no BPCR messages were delivered to the women and a maximum value of six indicating that the women received all six key messages. A similar type of content index was used by other recent studies [22, 23].

**Explanatory variables.** Religion, ethnicity, age, place of residence (urban and rural), educational level (no education, primary, secondary, and higher), husband's education (no education, primary, secondary, higher), and wealth status of women's household were considered from socioeconomic and demographic characteristics of women. The wealth index was divided into five categories: poorest, poorer, middle, richer, and richest. The wealth quintile of women's households in the EDHS is a composite indicator based on housing characteristics and ownership of household durable goods that was calculated using principal component analysis.

Obstetric characteristics like parity (nulliparous, primiparous, multiparous, and grand multiparous), gravidity, the total number of birth in the last five years, pregnancy status when she became pregnant (wanted, mistimed, unwanted), total children ever born, ever had a termination of pregnancy. Maternal health service-related characteristics like the frequency of Antenatal care visits, place of receiving ANC (home, public, private, and NGO), contraceptive use (Yes or No), the decision-making power on own health care (self-decision/joint decision with husband, husband alone, and other), covered by health insurance(Yes or No), and exposure to the newspaper, radio, and television (not at all, less than once a week or at least once a week) were considered. Furthermore, problems encountered by women in accessing medical help for themselves, such as distance to a health facility, obtaining permission to visit a health facility,

and obtaining the money required for treatment, were assessed and rated as a big problem or not a big problem. Some of the behavioral characteristics of the respondents like alcohol consumption and cigarette smoking were assessed with a "Yes" or "No" response.

## Statistical methods and data analysis

The necessary data from EDHS 2016 report were checked for consistency and missing values. STATA/SE version 16.0 was used for cleaning, recoding, variable generation, labeling, and analysis. The sample allocation to different regions, as well as urban and rural settings, was not proportional in the EDHS. As a result, sample weights were used to estimate proportions and frequencies to account for disproportionate sampling and non-response. The weighting procedure was thoroughly explained in the 2016 EDHS report [4]. Descriptive statistics were computed to describe the characteristics of the overall sample respondents (mothers) across a set of covariates.

The use of a multilevel modeling approach accounts for the EDHS data's hierarchical nature, as households were selected within EA clusters. There may be unobserved cluster characteristics influencing BPCR message uptake among women, such as the availability and accessibility of health services, cultural norms, and predominant health beliefs [4]. Thus multilevel mixed-effects models (cluster/region-specific random effects) were applied to identify the predictors. Accordingly, two different modes of analysis were implemented to estimate both the independent (fixed) and community-level (random) effect of the explanatory variables on our dependent variables:

i. First, to examine the relationship between each predictor and the first outcome variable(a receipt of at least one BPCR message), a multilevel bivariable logistic regression analysis was performed. In this analysis, variables with p-values less than 0.25 were candidates for a multilevel multivariable mixed-effect logistic regression analysis. Then a multilevel mixed-effects logistic regression analysis was run. In a multivariable multilevel mixed-effect logistic analysis, four models with the variables of interest were fitted, and the best-fitting model was chosen. Model-I is a null model, Model II is a model with only individual-level factors, Model III is a model with only community-level factors, and Model IV is a full model. The full model (Model IV) was fitted to examine the effect of individual and community-level predictors on the outcome variable at the same time. The adjusted odds ratio with the corresponding 95% confidence interval was computed and reported to demonstrate the strength of the association and its significance. Variables having a p-value <0.05 were considered as having a significant association with the outcome variable. The model comparison was done using deviance and the fourth model with the lowest deviance was selected as the best-fitted model

ii. To identify factors associated with the secondary outcome variable (a receipt of the recommended number of BPCR messages), a generalized linear model (GLM) with a multilevel mixed-effect negative binomial regression was run. Since the number of key BPCR messages received is a non-negative integer (count), most of the recent thinking in the field has used the Poisson regression model as a starting point [24, 25]. The most serious limitation of Poisson regression is that it assumes that the variance of the count response variable's distribution is equal to its mean, which is known as the assumption of equidispersion. If this assumption is breached, the Poisson regression model's estimates remain consistent but produce incorrect inferences about the parameters [26]. In the current case, the mean and the variance of the count outcome variable were 1.25 and 2.51, respectively. As a result of the assumption being violated, the data were over-dispersed, and a multilevel mixed-effect negative binomial regression model was fitted [25, 27]. Independent t-tests and analysis of variance (ANOVA) were used to determine whether there were statistically significant differences in the mean number

of BPCR messages across each categorical variable. Those variables with p-values less than 0.05 were eligible for a multilevel mixed-effect negative binomial regression using a generalized linear model (GLM) to identify the determinants of the number of BPCR messages uptake. During multivariable multilevel mixed-effect negative binomial regression, four models with the variables of interest were fitted, and the best-fitting model was chosen. Model-I is a null model, Model II is a model with only individual-level factors, Model III is a model with only community-level factors, and Model IV is a full model. The full model (Model IV) was fitted to examine the effect of individual and community-level predictors on the outcome variable at the same time. Finally, the incident rate ratio (IRR) with a 95% confidence interval was reported, and statistical significance was determined at a p-value less than 0.05.

## Model building and selection

Measures of random effect like Intra-class correlation coefficient (ICC), a proportional change in variance (PCV), and median odds ratio (MOR) were estimated. ICC explains the cluster variability, while MOR can quantify unexplained cluster variability (heterogeneity). In both cases (first and second outcome variables), the results of the random effects model showed the presence of variations of the random factor in the null model, indicating the existence of variation in the receipt of BPCR messages. Thus, to account for this variation, a multilevel mixed-effect logistic regression (for the first outcome variable) and a multi-level mixed-effect binomial regression (for the second outcome variable) model were considered for further analysis (Tables 5 and 6). Model IV had the lowest AIC value in both cases (primary and secondary outcome variables) and was selected as the best model fit for the data. Furthermore, as fitted models progressed from the empty model (Model-I) to Model-II, Model-III, and Model-IV, the value of the deviance (-2*log-likelihood) results consistently decreased, indicating that the fitted models were a better fit to the data.

## Ethical approval and consent to participate

Following registration with possible justification, ICF International granted permission to access the dataset used for this study. The retrieved data were only used for the registered research, and data were not shared with anyone other than the coresearchers. The information was kept private, and no attempt was made to identify any household or individual respondent.

## Results

### Sociodemographic characteristics of women

The Oromia region had the highest proportion of study participants 1,607.33(33.69%), followed by the SNNPR 1,114.51(23.36%), and the Amhara region 1,103.84(23.13%). The mean (±SD) age of study participants was 28.79(±6.61) years, of which nearly one-third (30.49%) of them belongs to the age group 25–29 years. The vast majority of respondents, 3896.31(81.66%) were rural residents. The proportion of women with no formal education (39.98%) and primary education (40.53%) was comparable.

 The highest (1.45) and lowest (0.81) mean scores of BPCR messages were reported by women aged 40–44 and 15–19 years, respectively. There was a statistically significant difference in the uptake of the number of BPCR messages across the educational level of women (p<0.001). Women with a higher educational level received the highest mean of BPCR messages (1.67). Women in the poorest household wealth index received the lowest mean number of BPCR messages (0.96), while those in the richest wealth index received the highest (1.57) (Table 1).

**Table 1. Sociodemographic characteristics of women in the reproductive age across magnitude and mean number of BPCR messages in Ethiopia, 2016.**

| Variable categories | Weighted frequency (%) | Received BPCR message | | Mean number of BPCR Items received (95%CI) | p-value |
|---|---|---|---|---|---|
| | | Yes (%) | No (%) | | |
| **Regions** | | | | | |
| Tigray | 485.61(10.18) | 374.79 (14.03) | 110.81(5.28) | 1.76 (1.63, 1.876) | <0.001[b] |
| Afar | 36.62(0.77) | 8.03 (0.30) | 28.59 (1.36) | 0.43 (0.30, 0.53) | |
| Amhara | 1,103.84(23.13) | 726.88(27.20) | 376.96(17.95) | 1.29 (1.16, 1.42) | |
| Oromia | 1,607.33(33.69) | 743.67(27.83) | 863.66(41.13) | 0.97 (0.85,1.08) | |
| Somali | 118.24(2.48) | 38.82(1.45) | 79.42 (3.78) | 0.62 (0.49, 0.75) | |
| Benishangul | 55.94(1.17) | 27.37(1.02) | 28.57 (1.36) | 1.16 (1.01, 1.33) | |
| SNNPR | 1,114.51(23.36) | 603.78(22.60) | 510.72(24.32) | 1.17(1.06, 1.27) | |
| Gambela | 15.10(0.32) | 6.47(0.24) | 8.63(0.41) | 0.92(0.77, 1.07) | |
| Harari | 13.30(0.28) | 8.08 (0.30) | 5.22 (0.25) | 1.21(1.06, 1.37) | |
| Addis Ababa | 191.91(4.02) | 120.74 (4.52) | 71.16 (3.30) | 1.70(1.52, 1.89) | |
| Diredawa | 29.07(0.61) | 13.24 (0.50) | 15.82(0.75) | 0.86(0.73, 0.98) | |
| **Current Age** | | | | | |
| 15–19 | 251.43 (5.27) | 121.40(4.54) | 130.03(6.19) | 0.81(0.66, 0.96) | 0.029 [b] |
| 20–24 | 984.90 (20.64) | 548.44(20.53) | 436.46(20.79) | 1.08(1.01, 1.17) | |
| 25–29 | 1,454.74 (30.49) | 808.78(30.27) | 645.96(30.77) | 1.05(0.98, 1.13) | |
| 30–34 | 1,041.32 (21.82) | 568.43(21.27) | 472.89(22.52) | 1.06(0.97, 1.14) | |
| 35–39 | 659.909 (13.83) | 390.69(14.62) | 269.21(12.82) | 1.18(1.07, 1.28) | |
| 40–44 | 275.59 (5.78) | 169.98(6.36) | 105.61(5.03) | 1.45 (1.16, 1.78) | |
| 45–49 | 103.58 (2.17) | 64.15(2.40) | 39.42(1.88) | 1.12(0.82, 1.40) | |
| **Current Marital status** | | | | | |
| In marital relation | 4,429.76(92.84) | 2,479.02(92.78) | 1,950.7(92.91) | 1.35 (1.18, 1.48) | 0.607[b] |
| Live with partner | 51.13(1.07) | 27.99(1.05) | 23.14(1.10) | 1.01(0.62, 1.38) | |
| Not in marital relation | 290.60(6.09) | 164.88(6.17) | 125.72(5.99) | 1.30 (1.13, 1.46) | |
| **Religion** | | | | | |
| Orthodox | 2,029.97(42.54) | 1,312.01(49.10) | 717.95 (34.19) | 1.52(1.45, 1.59) | <0.001[b] |
| Catholic | 41.96(0.88) | 26.09(0.98) | 15.87(0.76) | 1.53(0.82, 2.25) | |
| Protestant | 1,049.22(21.99) | 546.55(20.46) | 502.66(23.94) | 1.22(1.11, 1.33) | |
| Muslim | 1,572.8(32.96) | 759.66(28.43) | 813.21(38.73) | 0.97(0.91, 1.24) | |
| Traditional | 51.910(1.09) | 11.78(0.44) | 40.13(1.91) | 0.44(0.10, 0.77) | |
| Other | 25.55(0.54) | 15.79(0.59) | 9.76(0.46) | 1.19(0.60, 1.97) | |
| **Educational status** | | | | | |
| No education | 2,580.00(54.07) | 1,394.82 (52.20) | 1,185.18(56.45) | 1.08(1.03,1.15) | 0.001[b] |
| Primary | 1,576.92(33.05) | 881.59(33.00) | 695.32(33.12) | 1.29(1.21, 1.37) | |
| Secondary | 387.60(8.12) | 242.25(9.07) | 145.35(6.92) | 1.49(1.3, 1.63) | |
| Higher | 226.97(4.76) | 153.22(5.73) | 73.74(3.51) | 1.67(1.47, 1.87) | |
| **Husband's education** | | | | | |
| No education | 1,791.39(39.98) | 995.75(39.72) | 795.6(40.31) | 1.06(0.99,1.13) | 0.061[b] |
| Primary | 1,816.17(40.53) | 997.60(39.79) | 818.56(41.47) | 1.29(1.2, 1.37) | |
| Secondary | 513.11(11.45) | 281.69 (11.24) | 231.42(11.72) | 1.24(1.12, 1.36) | |
| Higher | 335.55(7.49) | 217.66(8.68) | 117.89(5.97) | 1.37(1.31, 1.62) | |
| Don't know | 24.68(0.55) | 14.3(0.57) | 10.37(0.53) | 1.12(0.79, 1.85) | |
| **Occupation** | | | | | |
| Unemployed | 2,420.83(50.74) | 1,247.06 (46.67) | 1,173.77 (55.90) | 1.15(1.08, 1.21) | 0.144[a] |
| Employed | 2,350.66(49.26) | 1,424.84 (53.33) | 925.82(44.10) | 1.33(1.27,1.40) | |
| **Residence** | | | | | |
| Urban | 875.18(18.34) | 517.09(19.35) | 358.09(17.06) | 1.40 (1.31,1.49) | 0.002[a] |

*(Continued)*

**Table 1.** (Continued)

| Variable categories | Weighted frequency (%) | Received BPCR message | | Mean number of BPCR Items received (95%CI) | p-value |
|---|---|---|---|---|---|
| | | Yes (%) | No (%) | | |
| Rural | 3,896.31(81.66) | 2,154.80(80.65) | 1,741.50(82.94) | 1.16(1.11,1.22) | |
| **Family size** | | | | | |
| <5memeber | 2,513.70(52.68) | 1,424.06(53.30) | 1,089.65 (51.90) | 1.28(1.22,1.34) | 0.147[a] |
| > = 5 | 2,257.78(47.32) | 1,247.84(46.70) | 1,009.95(48.10) | 1.18(1.12,1.22) | |
| **Wealth index combined** | | | | | |
| Poorest | 794.48(16.65) | 389.05 (14.56) | 405.43 (19.31) | 0.96(0.87.1.05) | <0.001[b] |
| Poorer | 935.41(19.60) | 467.053(17.48) | 468.36(22.31) | 1.10(0.99, 1.23) | |
| Middle | 996.14(20.88) | 564.78(21.14) | 431.37(20.55) | 1.21 (1.10, 1.32) | |
| Richer | 966.66(20.26) | 568.44 (21.27) | 398.21(18.97) | 1.30(1.19,1.42) | |
| Richest | 1,078.79(22.61) | 682.57 (25.55) | 396.22(18.87) | 1.57(1.39, 1.86) | |

[a]p-values are based on an independent t-test for testing variation in means across covariate

[b]p-values are based on the Analysis of variance(ANOVA) for testing variation in means across covariate

## Obstetric characteristics of the respondents

This study included a total of 4,771.49 weighted women who responded to a question if they received a birth preparedness and complication readiness plan (BPCRP) message during any of their prenatal visits within the five years preceding the survey. The majority of respondents (46.97%) were multiparous (women with 2–4 living children), followed by grand multiparous (women with five or more living children) (27.38%). More than three-quarters (75.91%) of women reported that their pregnancy was planned, and 439.72(9.22%) experienced termination of pregnancy. More than half (51.52%) of respondents got at least four antenatal visits.

The mean number of BPCR messages received significantly increased as the frequency of ANC visits increased. Women with at least four ANC visits got the highest (1.68) mean of BPCR messages, while women who only had one ANC visit had the lowest (0.48) mean score. A significantly high mean number of BPCR messages was reported among those women who gave birth at health facilities (1.45) (Table 2).

## Individual and health system-related characteristics of respondents

In terms of media exposure, only 2.59%, 17.12%, and 13.76% of women reported that they read a newspaper or magazine, listened to the radio, and watched television at least once a week, respectively. Regarding decision-making power, roughly two-thirds (64.60%) of women reported that both women and husbands/partners jointly decide on accessing health care. Distance to health facilities and getting permission to seek medical care were big problems for 50.33% and 29.81% of women, respectively. Women who listened to the radio at least once a week had the highest mean number (1.59) of BPCR messages received, while women who never listened to the radio had the lowest mean number (1.12). Similarly, women who watched television at least once a week got the highest mean number of BPCR messages (1.55). Women who faced the problem of distance to health facilities and obtaining permission experienced a significant decrement in the average number of BPCR messages received (Table 3).

## The overall uptake of WHO-recommended BPCR messages

More than half of the women, 2,673.17 (56.02%), received at least one BPCR plan message. Almost a quarter (24.39%) received only one BPCR message, while 19.35% received at least

**Table 2. Obstetric characteristics of women in the reproductive age group across the overall uptake and mean number of BPCR in Ethiopia, 2016.**

| Variable categories | Weighted frequency (percentage) | Received BPCR message | | Mean number of BPCR Items received (95%CI) | P-value |
|---|---|---|---|---|---|
| | | Yes (%) | No (%) | | |
| **Parity** | | | | | |
| Nulliparous | 30.407(0.64) | 17.780(0.67) | 12.63(0.60) | 0.89(0.48, 1.30) | 0.190[b] |
| Primiparous | 1,193.12(25.01) | 690.59(25.85) | 502.53(23.93) | 1.29(1.20, 1.39) | |
| Multiparous | 2,241.39(46.97) | 1,227.59 (45.94) | 1,013.79 (48.29) | 1.23(1.17, 1.29) | |
| Grand multiparous | 1,306.58(27.38) | 735.93(27.54) | 570.64(27.18) | 1.19(1.10, 1.28) | |
| **Birth in the last five years** | | | | | |
| One | 3,121.61(65.42) | 1,771.01 (66.28) | 1,350.59 (64.33) | 1.30(1.24, 1.35) | 0.013 [b] |
| Two | 1,432.35(30.02) | 809.40(30.29) | 622.94(29.67) | 1.17(1.09, 1.25) | |
| More than two | 217.53(4.56) | 91.47(3.42) | 126.06(6.00) | 0.87(0.69, 1.05) | |
| **Place of delivery** | | | | | |
| Home delivery | 2,511.47(52.63) | 1,222.97 (45.77) | 1,288.5(61.37) | 0.97(0.91, 1.03) | <0.001[a] |
| Facility delivery | 2,260.02(47.37) | 1,448.92 (54.23) | 811.10(38.63) | 1.45(1.38, 1.51) | |
| **Pregnancy status when she became pregnant** | | | | | |
| Wanted | 3,621.84(75.91) | 1,969.53 (73.71) | 1,652.30 (78.70) | 1.40(1.19, 1.61) | 0.092[b] |
| Mistimed | 825.87(17.31) | 507.11(18.98) | 318.76(15.18) | 1.33(1.21, 1.45) | |
| Unwanted | 323.78(6.79) | 195.25(7.31) | 128.53(6.12) | 1.20(1.16, 1.26) | |
| **Total children ever born** | | | | | |
| One | 1,122.68(23.53) | 648.94 (24.29) | 473.74(22.56) | 1.13(1.19, 1.38) | 0.474[b] |
| 2–5 | 2,575.92(53.99) | 1,419.38 (53.12) | 1,156.54 (55.08) | 1.23(1.16, 1.29) | |
| > = 6 | 1,072.89(22.49) | 603.57(22.59) | 469.32(22.35) | 1.19(1.09, 1.29) | |
| Ever had a termination of pregnancy | | | | | |
| No | 4,331.77(90.78) | 2,409.12 (90.17) | 1,922.66 (91.57) | 1.22(1.17, 1.27) | 0.170[a] |
| Yes | 439.72(9.22) | 262.78(9.83) | 176.94(8.43) | 1.35(1.19, 1.50) | |
| Contraceptive utilization | | | | | |
| Non users | 2,773.90(58.13) | 1,478.01 (55.32) | 1,295.89 (61.72) | 1.14(1.08, 1.19) | 0.079[a] |
| Users | 1,997.59(41.87) | 1,193.88 (44.68) | 803.71(38.28) | 1.39(1.32, 1.47) | |
| Frequency of ANC | | | | | |
| 1 visit | 313.38(6.57) | 107.21(4.01) | 206.17(9.82) | 0.48(0.38, 0.58) | <0.001[b] |
| 2 visits | 580.72(12.17) | 257.93(9.65) | 322.79(15.37) | 0.86(0.74, 0.98) | |
| 3 visits | 1,418.89(29.74) | 766.18 (28.68) | 652.71(31.09) | 1.23(1.08, 1.15) | |
| ≥4 visits | 2,458.48 (51.52) | 1,540.57 (57.66) | 917.91(43.72) | 1.68(1.41, 1.84) | |

[a] p-values are based on an independent t-test for testing variation among means

[b] p-values are based on the Analysis of variance(ANOVA) for testing variation among means

three of the six WHO-recommended messages. The number of BPCR messages received ranged from 0 to 6, with a mean and variance of 1.25 and 2.51, respectively, indicating that over-dispersion did exist (Fig 1).

**Table 3. The distribution of individual and behavioral characteristics of women in the reproductive age group across the overall uptake and mean number of BPCR messages in Ethiopia, 2016.**

| Variables | Weighted frequency (percentage) | Received BPCR message | | Mean number contents of ANC (95% CI) | P-value |
|---|---|---|---|---|---|
| | | Yes (%) | No (%) | | |
| **Reading newspaper** | | | | | |
| Not at all | 4,294.71(90.01) | 2,356.92 (88.21) | 1,937.78 (92.29) | 1.18(1.13, 1.23) | 0.052[b] |
| Less than once a week | 353.23(7.40) | 236.28(8.84) | 116.94(5.57) | 1.62(1.47, 1.78) | |
| At least once a week | 123.55(2.59) | 78.68(2.95) | 44.86(2.14) | 1.67(1.37, 1.97) | |
| **Listening to a radio** | | | | | |
| Not at all | 3,161.66(66.26) | 1,749.10 (65.46) | 1,412.56 (67.28) | 1.12(1.07, 1.18) | <0.001[b] |
| Less than once a week | 792.74(16.61) | 451.15(16.88) | 341.59(16.27) | 1.34(1.23, 1.45) | |
| At least once a week | 817.08(17.12) | 471.64(17.65) | 345.4(16.45) | 1.59(1.47, 1.72) | |
| **Watching television** | | | | | |
| Not at all | 3,563.66(74.69) | 1,949.85 (72.98) | 1,613.82 (76.86) | 1.14(1.09, 1.19) | <0.001[b] |
| Less than once a week | 551.3918(11.56) | 320.36(11.99) | 231.03(11.00) | 1.22(1.09, 1.34) | |
| At least once a week | 656.44(13.76) | 401.69(15.03) | 254.75(12.13) | 1.55(1.44, 1.66) | |
| **Had a mobile phone** | | | | | |
| No | 3,605.3(75.56) | 1,959.47 (73.34) | 1,645.84 (78.39) | 1.14(1.08, 1.19) | 0.001[a] |
| Yes | 1,166.18(24.44) | 712.42(26.66) | 453.76(21.61) | 1.42(1.34, 1.50) | |
| **Ever taken a drink that contains alcohol?** | | | | | |
| No | 3,073.98(64.42) | 1,577.09 (59.03) | 1,496.89 (71.29) | 1.09(1.04, 1.15) | 0.126[a] |
| Yes | 1,697.52(35.58) | 1,094.80 (40.97) | 602.71(28.71) | 1.22(1.14, 1.38) | |
| **Smoke cigarette** | | | | | |
| No | 4,752.53(99.60) | 2,666.44 (99.80) | 2,086.09 (99.36) | 1.24(1.19,1.28) | 0.212[a] |
| Yes | 18.96(0.40) | 5.46(0.20) | 13.51(0.64) | 0.78(0.32, 1.23) | |
| **Who decides on healthcare** | | | | | |
| Respondent alone | 679.39(14.24) | 403.34(15.10) | 276.05(13.15) | 1.32(1.16,1.49) | 0.300[b] |
| Respondent and husband | 3,082.45(64.60) | 1,712.54 (64.09) | 1,369.91 (65.25) | 1.24(1.13, 1.34) | |
| Husband/partner alone | 709.30(14.87) | 381.67(14.28) | 327.6(15.60) | 1.28(1.22, 1.35) | |
| Someone else | 300.34(6.29) | 174.35(6.53) | 125.99(6.0) | 1.03(0.92, 1.14) | |
| **Distance to a health facility** | | | | | |
| Big problem | 2,401.42(50.33) | 1,233.92 (46.18) | 1,167.50 (55.61) | 1.10(1.04, 1.17) | 0.019[a] |
| Not a big problem | 2,370.07(49.67) | 1,437.97 (53.82) | 932.09(44.39) | 1.34(1.28,1.40) | |
| **Getting permission to go to a health facility** | | | | | |
| Big problem | 1,422.55(29.81) | 1,233.92 (46.18) | 1,167.50 (55.61) | 1.04(0.95, 1.12) | 0.007[a] |
| Not a big problem | 3,348.94(70.19) | 1,437.97 (53.82) | 932.09(44.39) | 1.34(1.25, 1.46) | |
| **Getting money needed for treatment** | | | | | |
| Big problem | 2,560.56(53.66) | 1,353.73 (50.67) | 1,206.84 (57.48) | 1.15(1.09, 1.22) | 0.625[a] |

(*Continued*)

**Table 3.** (Continued)

| Variables | Weighted frequency (percentage) | Received BPCR message | | Mean number contents of ANC (95% CI) | P-value |
|---|---|---|---|---|---|
| | | Yes (%) | No (%) | | |
| Not a big problem | 2,210.93(46.34) | 1,318.17 (49.33) | 892.76(42.52) | 1.32(1.25, 1.38) | |
| **Covered by health insurance** | | | | | |
| No | 4,530.27(94.94) | 2,488.59 (93.14) | 2,041.68 (97.24) | 1.23(1.17,1.26) | 0.011[a] |
| Yes | 241.22(5.06) | 183.30(6.86) | 57.92(2.76) | 1.70(1.48, 1.93) | |

[a] p-values are based on an independent t-test for testing variation among means

[b] p-values are based on the Analysis of variance(ANOVA) for testing variation among mean

**Items BPCR messages received by mothers.**  The 2016 EDHS collected data on six essential BPCR messages delivered during prenatal visits. Those key messages were about determining the place of birth, obtaining necessary supplies for childbirth, preparing emergency transportation, saving money for emergency expenses, identifying companions during labor and childbirth, and securing potential blood donors. Only 140(2.94%) women received all six messages, according to the study. In terms of individual items, the commonest message received by nearly half (48.42%) of mothers was about identifying a place of birth, followed by securing the necessary supplies for childbirth (28.85%) (Fig 2).

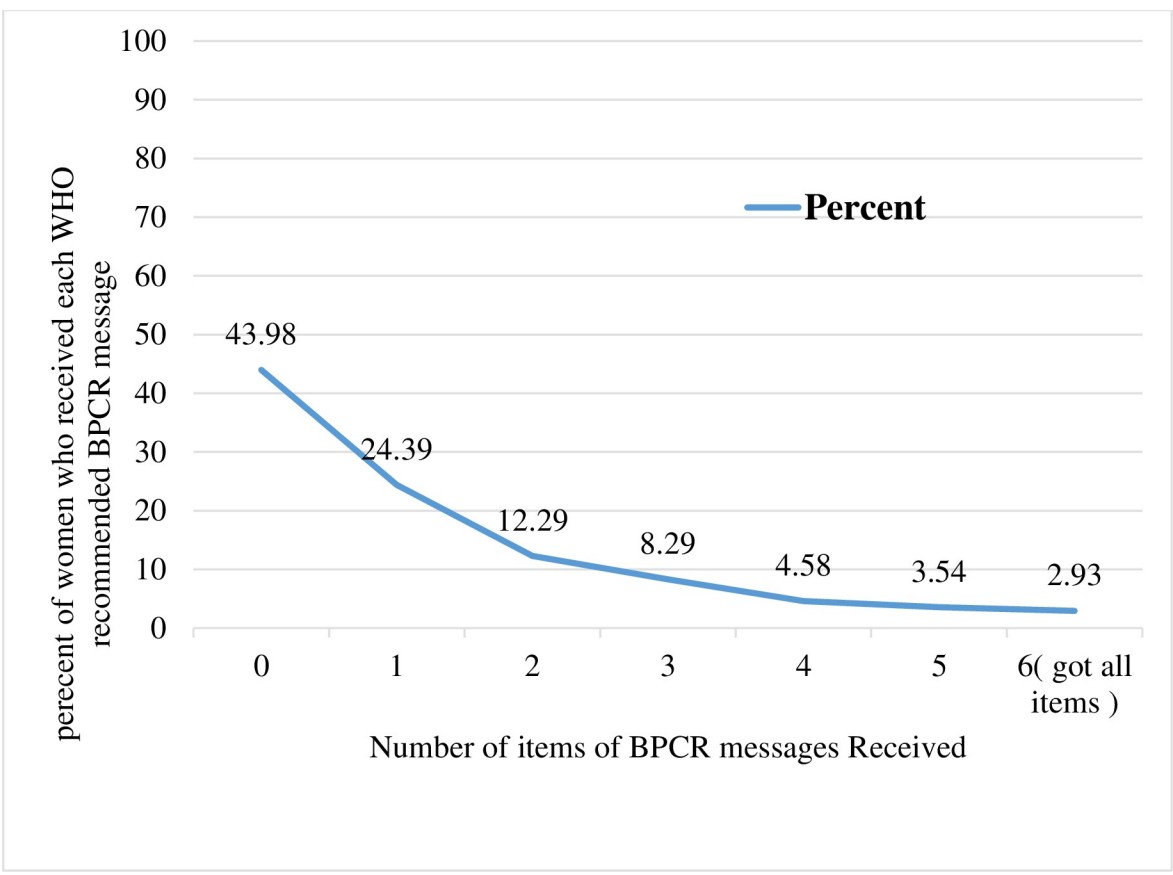

**Fig 1. Number of BPCR messages received by women during their recent pregnancy in Ethiopia, 2016.**

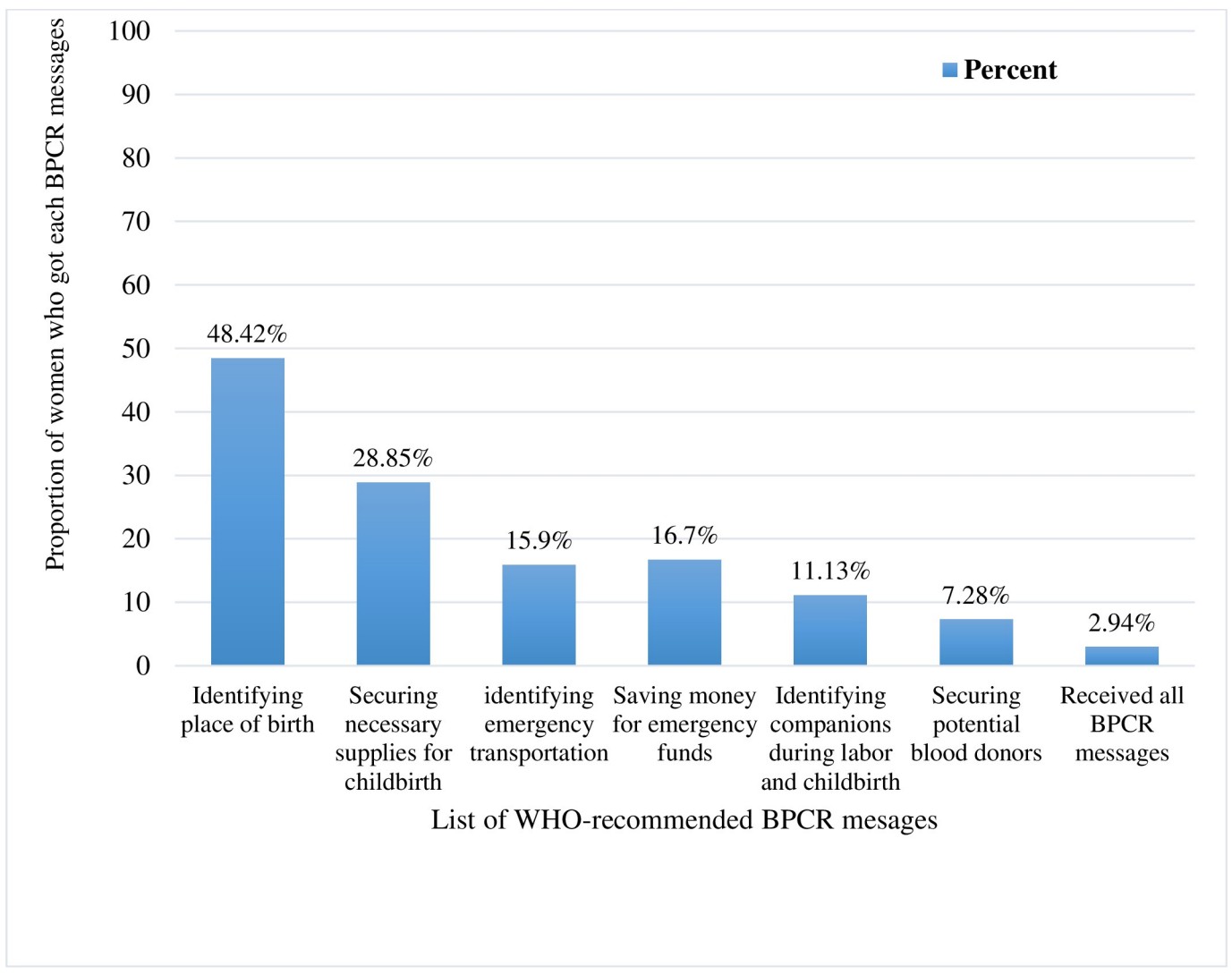

**Fig 2. Percentage distribution of the number of items of BPCR messages received by pregnant women during their recent pregnancy in Ethiopia, EDHS 2016.**

### Factors affecting the uptake of BPCR message: A multilevel mixed-effects logistic regression analysis

In bivariable multilevel logistic regression, 15 variables namely region, religion, respondents' educational status, being employed, wealth index, number of births in the last five years, experiencing termination of pregnancy, contraceptive uptake, frequency of ANC, reading a newspaper, having a mobile phone, distance to a health facility, having permission to seek medical care, having money for medical care, and enrollment in health insurance were associated with uptake of BPCR at p-value <0.25. All the above-mentioned variables were entered into a multilevel mixed-effect multivariable logistic regression (Table 4).

In the multilevel multivariable mixed-effects logistic regression analysis, the region of respondents, employment status, the combined household wealth index, frequency of ANC visits, ease of obtaining permission to seek medical care, and enrollment in health insurance schemes were found to be significantly associated with the uptake of BPCR message.

**Table 4. Results of a multilevel bivariable logistic regression analysis to identify the factors associated the uptake of BPCR messages among pregnant women in Ethiopia, 2016.**

| Variable categories | Received BPCR messages | | COR (95%CI) |
|---|---|---|---|
| | Yes (%) | No (%) | |
| **Regions** | | | |
| Tigray | 374.79(77.18) | 110.81(22.82) | 12.05(8.4,17.17) |
| Amhara | 726.88(65.85) | 376.96(34.15) | 6.86(4.80, 9.82) |
| Oromia | 743.67(46.26) | 863.66(53.74) | 3.06(2.16,4.36) |
| Somali | 38.82(48.92) | 79.42(67.17) | 1.74(1.15, 2.63) |
| Benishangul | 27.37(48.92) | 28.57(51.08) | 3.41(2.36,4.93) |
| SNNPR | 603.78(54.17) | 510.72(45.83) | 4.21(2.99, 5.94) |
| Gambella | 6.47(42.85) | 8.63(57.15) | 2.67(1.81, 3.93) |
| Harari | 8.08 (60.75) | 5.22(39.25) | 5.51(3.77, 8.07) |
| Addis Ababa | 120.74 (62.92) | 71.16(37.08) | 6.04(4.15, 8.79) |
| Diredawa | 13.24(45.56) | 15.82(54.44) | 2.98(2.04, 4.36) |
| Afar | 8.03(21.93) | 28.59(78.07) | 1 |
| **Religion** | | | |
| Orthodox | 1,312.01(64.63) | 717.95 (35.37) | 9.6(2.47, 27.37) |
| Catholic | 26.09(62.18) | 15.87(37.82) | 6.13(1.14, 22.95) |
| Protestant | 546.55(52.09) | 502.66(47.91) | 7.14(1.07, 28.75) |
| Muslim | 759.66(48.29) | 813.21(51.71) | 4.84(1.23, 19.05) |
| Other | 15.79(61.80) | 9.76(38.20) | 5.51(0.92, 22.97) |
| Traditional | 11.78(22.69) | 40.13(77.31) | 1 |
| **Educational status** | | | |
| Higher | 153.22(67.51) | 73.74(32.49) | 2.07(1.19, 3.59) |
| Secondary | 242.25(62.50) | 145.35(37.50) | 1.40(0.97, 2.02) |
| Primary | 881.59(55.90) | 695.32(44.10) | 1.07(0.89, 1.30) |
| No education | 1,394.82(54.06) | 1,185.18(45.94) | 1 |
| **Occupation** | | | |
| Employed | 1,424.84(60.61) | 925.82(39.39) | 1.37(1.09, 1.71) |
| Unemployed | 1,247.06(51.51) | 1,173.77(48.49) | 1 |
| **Wealth index combined** | | | |
| Richest | 682.57 (63.27) | 396.22(36.73) | 1.94(1.34, 2.80) |
| Richer | 568.44 (58.80) | 398.21(41.20) | 1.43(1.02, 2.01) |
| Middle | 564.78(56.67) | 431.37(53.33) | 1.36(1.03, 1.80) |
| Poorer | 467.05(49.92) | 468.36(50.08) | 1.04(0.79, 1.37) |
| Poorest | 389.05(48.97) | 405.43(51.03) | 1 |
| **Birth in the last five years** | | | |
| One | 1,771.01(56.73) | 1,350.59(43.27) | 1.76(1.09, 2.85) |
| Two | 809.40(56.50) | 622.94(43.5) | 1.97(1.22, 3.18) |
| More than two | 91.47(42.05) | 126.06(57.95) | 1 |
| **Ever faced termination of pregnancy** | | | |
| Yes | 262.78(59.76) | 176.94(40.24) | 1.18(0.89, 1.58) |
| No | 2,409.12(55.61) | 1,922.66(44.29) | 1 |
| **Contraceptive uptake** | | | |
| Users | 1,193.88(59.76) | 803.71(40.24) | 1.22(0.99, 1.50) |
| Non-users | 1,478.01(53.28) | 1,295.89(46.72) | 1 |
| **Frequency of ANC** | | | |
| ≥4 visits | 1,540.57(62.66) | 917.91(37.44) | 3.48(2.16, 5.63) |

(*Continued*)

**Table 4.** (Continued)

| Variable categories | Received BPCR messages | | COR (95%CI) |
|---|---|---|---|
| | Yes (%) | No (%) | |
| 3 visits | 766.18 (53.99) | 652.71(46.01) | 2.30 (1.36, 3.88) |
| 2 visits | 257.93(44.41) | 322.79(55.59) | 1.72 (1.04, 2.84) |
| 1 visit | 107.21(34.21) | 206.17(65.79) | 1 |
| **Reading newspaper** | | | |
| At least once a week | 78.68(63.09) | 44.86(36.91) | 1.67(0.84, 3.33) |
| Less than once a week | 236.28(66.89) | 116.94(32.11) | 1.64 (1.15, 2.36) |
| Not at all | 2,356.92(54.88) | 1,937.78(45.12) | 1 |
| **Had a mobile phone** | | | |
| Yes | 712.42(61.09) | 453.76(38.91) | 1.32(1.08, 1.61) |
| No | 1,959.47(54.35) | 1,645.84(45.65) | 1 |
| **Distance to a health facility** | | | |
| Big problem | 1,233.92(51.38) | 1,167.50(49.62) | 0.77(0.61, 0.97) |
| Not a big problem | 1,437.97(60.67) | 932.09(39.33) | 1 |
| **Getting permission to go to a health facility** | | | |
| Big problem | 668.48(46.99) | 754.06(53.01) | 0.59(0.49, 0.72) |
| Not a big problem | 2,003.40() | 1,345.53() | 1 |
| **Getting money needed for treatment** | | | |
| Big problem | 1,353.73(52.86) | 1,206.84(47.14) | 0.76(0.64, 0.90) |
| Not a big problem | 1,318.17(57.29) | 892.76(42.71) | 1 |
| **Covered by health insurance** | | | |
| No | 2,488.59(54.93) | 2,041.68(45.07) | 0.38(0.26, 0.57) |
| Yes | 183.30(75.99) | 57.92(24.01) | 1 |

**Key:** 1: Reference category; COR = Crude odds ratio

Respondents from Tigray, Amhara, and Addis Ababa had 10.28 (AOR = 9.41, 95% CI: 5.84, 15.18), 6.13 (AOR: 6.13; 95% CI: 3.82, 9.54), and 4.92 (AOR: 4.92; 95% CI: 3.23,7.49) times higher odds of receiving BPCR messages than women in Afar. Employed women had a 33% higher chance of receiving BPCR messages as compared to their counterparts (AOR: 1.33; 95% CI: 1.12, 1.79). Women in the richest wealth quintiles were nearly 2 times more likely to receive BPCR messages as compared to their counterparts who lived in the poorest wealth quintiles (AOR = 1.86; 95% CI: 1.24, 2.80). The frequency of ANC visits was also identified as a significant predictor for the uptake of BPCR messages. Those women who received 4 and more ANC visits had a 3.21 times higher chance of receiving BPCR messages (AOR = 3.21; 95% CI: 1.96, 5.24). On the other hand, having difficulty obtaining permission to seek medical care and not being enrolled in community health insurance schemes were found to reduce BPCR message uptake. Women who had difficulty in obtaining permission to seek medical care were 37% less likely to get BPCR messages (AOR = 0.63; 95% CI: 0.53, 0.85). Similarly, those respondents who didn't enroll in a health insurance scheme were 35% less likely to receive BPCR messages as compared to their counterparts (AOR = 0.65; 95% CI: 0.42, 0.98) (Table 5).

## Differentials of the receipt of WHO-recommended BPCR messages: A multilevel mixed-effect negative binomial regression analysis

The results of the bivariate analysis(one-way ANOVA and independent t-test) using the mean of the number of items of ANC services across a set of explanatory variables indicate that the

**Table 5. Results of a multilevel mixed-effect multivariable logistic regression analysis to identify the factors affecting the uptake of BPCR messages among pregnant women in Ethiopia, 2016.**

| Variable categories | Model I(nul model) | Model II (individual-level factors) | Model III (community-level factors) | Model-IV (full model) |
|---|---|---|---|---|
| | AOR (95% CI) | AOR (95% CI) | AOR (95% CI) | AOR(95% CI) |
| **Religion** | | | | |
| Orthodox | | 1.38(0.37, 5.14) | | 2.79(0 .98, 7.95) |
| Catholic | | 0.93(0.19, 4.31) | | 3.31(0.77, 14.24) |
| Protestant | | 0.88(0.23, 3.28) | | 3.31 (0.74, 9.64) |
| Muslim | | 0.75(0.20, 2.78) | | 2.75(0.97, 7.78) |
| Other | | 0.84(0.16, 4.23) | | 2.76(0.42, 18.02) |
| Traditional | | 1 | | 1 |
| **Educational status** | | | | |
| Higher | | 1.27(0.71, 1.41) | | 1.39 (0.73, 2.63) |
| Secondary | | 1.15(0.83, 1.41) | | 1.02(0.67, 1.56) |
| Primary | | 1.04(0.88, 1.22) | | 1.06(0.85, 1.31) |
| No education | | 1 | | 1 |
| **Occupation** | | | | |
| Employed | | 1.12(0.97, 1.27) | | 1.33(1.12, 1.79)* |
| Unemployed | | 1 | | 1 |
| **Wealth index combined** | | | | |
| Richest | | 2.33(1.43, 3.73)** | | 1.86(1.24, 2.80)* |
| Richer | | 1.61(1.25, 2.06)** | | 1.19(0.82, 1.73) |
| Middle | | 1.45(1.15, 1.84)** | | 1.20(0.81, 1.77) |
| Poorer | | 0.98(0.85, 1.33) | | 0.84(0.60, 1.17) |
| Poorest | | 1 | | 1 |
| **Birth in the last five years** | | | | |
| One | | 1.39(1.03, 1.89)** | | 1.34(0.811, 2.24) |
| Two | | 1.54(1.13, 2.10)** | | 1.72 (0.98, 2.86) |
| More than two | | 1 | | 1 |
| **Ever faced termination of pregnancy** | | | | |
| Yes | | 1.12(0.89, 1.40) | | 1.21(0.89, 1.64) |
| No | | 1 | | 1 |
| **Contraceptive utilization** | | | | |
| Users | | 1.07(0.92, 1.24) | | 1.06(0.87, 1.29) |
| Non-users | | 1 | | 1 |
| **Frequency of ANC** | | | | |
| ≥4 visits | | 3.33(2.49, 4.45)** | | 3.21(1.96, 5.24) * |
| 3 visits | | 2.35(1.74, 3.18)** | | 2.26(1.33, 3.83) * |
| 2 visits | | 1.56(1.12, 2.18)** | | 1.76(1.06, 2.91) * |
| 1 visit | | 1 | | 1 |
| **Reading newspaper** | | | | |
| At least once a week | | 1.27(1.07, 1.65)** | | 1.32(0.67, 2.60) |
| < once a week | | 1.13(0.88, 1.50) | | 1.24(0.85, 1.79) |
| Not at all | | 1 | | 1 |
| **Had a mobile phone** | | | | |
| Yes | | 1.43(0.79, 1.88) | | 1.23(0.77, 1.38) |
| No | | 1 | | 1 |
| **Distance to a health facility** | | | | |
| Big problem | | 0.86(0.74,1.02) | | 0.95(0.77, 1.19) |

*(Continued)*

**Table 5.** (Continued)

| Variable categories | Model I(nul model) | Model II (individual-level factors) | Model III (community-level factors) | Model-IV (full model) |
|---|---|---|---|---|
| | AOR (95% CI) | AOR (95% CI) | AOR (95% CI) | AOR(95% CI) |
| Not a big problem | | 1 | | 1 |
| **Getting permission to go to a health facility** | | | | |
| Big problem | | 0.78(0.59, 0.92) | | 0.63(0.53, 0.85)* |
| Not a big problem | | 1 | | 1 |
| **Getting money needed for treatment** | | | | |
| Big problem | | 0.91(0.75, 1.09) | | 1.11(0.82, 1.42) |
| Not a big problem | | 1 | | 1 |
| **Covered by health insurance** | | | | |
| No | | 0.38(0.26, 0.57) | | 0.65(0.42, 0.98)* |
| Yes | | | 1 | |
| **Regions** | | | | |
| Tigray | | | 13.43(8.70, 20.74) | 9.41(5.84, 15.18)* |
| Amhara | | | 8.16(5.27, 12.62) | 6.13(3.82, 9.54)* |
| Oromia | | | 3.37(2.19, 5.16) | 2.97(1.97, 4.47)* |
| Somali | | | 1.43(0.89, 2.29) | 1.97(1.27, 3.06)* |
| Benishangul | | | 4.01(2.54, 6.30) | 2.89(1.89, 4.41)* |
| SNNPR | | | 4.77(3.12, 7.28) | 3.59(2.32, 5.54)* |
| Gambella | | | 2.56(1.60, 4.09) | 1.36(1.36, 3.46) |
| Harari | | | 6.11(3.78, 9.86) | 4.92(3.23,7.49) |
| Addis Ababa | | | 5.51(3.39, 8.91) | 3.37(2.08, 5.46) |
| Diredawa | | | 2.83(1.76, 4.57) | 2.41(1.56, 3.72) |
| Afar | | | 1 | 1 |
| **Residence** | | | | |
| Urban | | | 1.54 (1.11, 1.96) | 1.12(0.86, 1.83) |
| Rural | | | 1 | 1 |
| **Random effects** | | | | |
| Variance | 0.93 | 0.56 | 0.39 | 0.34 |
| ICC | 0.22 | 0.13 | 0.14 | 0.12 |
| AIC | 6221.46 | 6027.54 | 6009.18 | 5699.65 |
| BIC | 6234.38 | 6214.82 | 6093.141 | 5757.97 |
| MOR | 2.50 | 2.05 | 1.83 | 1.76 |
| PCV | Reference | 0.39 | 0.58 | 0.63 |
| **Model fitness** | | | | |
| Log-likelihood | -3108.73 | -2984.77 | -2991.59 | -2909.82 |
| Deviance | 6217.46 | 5969.54 | 5983.18 | 5819.64 |

**Key:** 1: Reference category; AOR = Adjusted odds ratio

** Statistically significant at p-value <0.05

region where the residents lived in, religion, age of the respondent, educational status, residence, wealth index, the total number of children born in the last five years, reading a newspaper, listening to radio, watching television, having a mobile phone, distance to a health facility, having permission to go to a health facility, and being enrolled in health insurance as factors affecting the mean uptake of BPCR messages. The foregoing bivariate analyses present unadjusted effects of the explanatory variables on the outcome of interest. Thus, a multilevel mixed-

effect negative binomial regression model was fitted to determine the adjusted or net effect of an explanatory variable on the receipt of WHO-recommended BPCR items. The estimated incidence rate ratio (IRR) calculated the net effect of an explanatory variable after controlling for the effects of all other explanatory variables. Accordingly, region, having permission to go to a health facility, and frequency of ANC were identified as significant predictors of receiving the recommended number of key BPCR messages.

As compared to women from the Afar region, the number of getting adequate BPCR messages was 3.76 (IRR = 3.76, 95% CI: 2.65, 5.33) and 2.91 (IRR = 2.91, 95% CI: 2.04, 4.14) times higher among women who lived in Tigrai and Amhara regions, respectively. The frequency of ANC visits has a significant positive relationship with receiving an adequate number of BPCR messages, as the frequency of ANC visits increases the number of BPCR messages received also increases. Mothers who had four or more ANC visits were 2.78 times more likely to receive the recommended messages of BPCR services (IRR = 2.78; 95% CI: 2.09, 3.71) than mothers who had only one ANC visit. Women who hadn't faced a problem in getting permission to visit a health facility were 29% (IRR = 1.29; 95% CI: 1.028, 1.38) more likely to have a higher number of BPCR messages during their pregnancy (Table 6).

## Discussion

Because complications during pregnancy, intrapartum, and postpartum are unpredictable, every woman should be aware of and prepared for the key danger signs of obstetric complications during those critical periods [28, 29]. Birth preparedness and complication readiness (BPCR) is a critical component of ANC that aims to reduce unnecessary delays in seeking emergency obstetric care and thus improve maternal and neonatal outcomes [29, 30].

This study aimed at assessing the magnitude of BPCR message uptake among women in Ethiopia by using the EDHS report of 2016. According to the analysis, more than half of the women (56.02%) with a birth in the five years preceding the survey received at least one BPCR message during pregnancy. The finding was in tandem with studies conducted in Kenya (56.1%) [19], and Tanzania(58.2%) [20]. On the other hand, the finding was higher than reports of primary studies conducted in India(47.8%) [31], Kenya(11.4%) [32], Uganda (22.3%) [33], Rwanda(35%) [34], Nigeria(25,5%) [35], Cameroon (18.8%) [21], and Tanzania (12%) [36]. However, the finding was lower than two studies conducted in Nigeria(87.4%), (82.1%) [37, 38], and Kenya (70.5%) [39]. The disparities could be attributed to differences in study population coverage, as the majority of the studies we used to compare this finding were small-scale studies conducted at the district level. The present finding indicated that service uptake was insufficient in comparison to the WHO standard that every pregnant woman receives a BPCR message during her pregnancy [12, 29], implying the need for a concerted effort from stakeholders and healthcare providers to improve BPCR. As Ethiopia is one of the Sub-Saharan African countries with slow progress in maternal health and a high maternal mortality rate [40, 41], the implementation of these BPCR packages need to be strengthened at maternal health service delivery points.

Variables namely, the frequency of ANC visits and the ease of obtaining permission to seek medical care, and enrollment in health insurance schemes were identified as significant predictors of overall uptake and number of BPCR messages. Employment status and the household wealth index, on the other hand, were found to be significantly associated with the uptake of at least one BPCR message.

The analysis indicates that there was a significant positive association between the frequency of ANC visits and the uptake of BPCR messages and a higher number of BPCR messages. Those women who received four and more ANC visits had a 2.95 times higher chance of

**Table 6. Results of a multivariable mixed-effect negative binomial regression to identify the determinants of uptake of items of BPCR during ANC visit, Ethiopia 2016.**

| Variable categories | Model I (nul model) | Model II (individual-level factors) | Model III (community-level factors) | Model-IV (full model) |
|---|---|---|---|---|
| | | IRR(95%CI) | IRR(95%CI) | IRR(95%CI) |
| **Current Age** | | | | |
| 45–49 | | 1.19(0.98, 1.45) | | 1.18(0.77, 1.79) |
| 40–44 | | 1.26(1.03, 1.53) | | 1.19(0.88, 1.60) |
| 35–39 | | 1.36(1.11, 1.67) | | 1.23(0.92, 1.64) |
| 30–34 | | 1.32(1.06, 1.72) | | 1.22 (0.92, 1.62) |
| 25–29 | | 1.51(1.08, 2.09) | | 1.11(0.84, 1.46) |
| 20–24 | | 1.31(1.08, 2.09) | | 1.16(0.89, 1.52) |
| 15–19 | | 1 | | 1 |
| **Religion** | | | | |
| Orthodox | | 1.88(0.77, 4.59) | | 2.21(0.79, 6.15) |
| Catholic | | 1.39(0.57, 3.41) | | 2.37(0.78, 7.19) |
| Protestant | | 1.67(0.68, 4.08) | | 2.37(0.85, 6.59) |
| Muslim | | 1.39(0.57, 3.41) | | 2.19(0.79, 6.06) |
| Other | | 1.49(0.52, 4.26) | | 2.15(0.82, 8.50) |
| Traditional | | 1 | | 1 |
| **Educational status** | | | | |
| Higher | | 1.14(0.94, 1.38) | | 1.25(0.95, 1.63) |
| Secondary | | 1.10(0.94, 1.28) | | 1.08(0.87, 1.34) |
| Primary | | 1.08(0.98, 1.19) | | 1.03(0.90, 1.18) |
| No formal education | | 1 | | 1 |
| **Wealth index combined** | | | | |
| Richest | | 1.14(0.94, 1.38) | | 1.24(0.99, 1.54) |
| Richer | | 1.11(0.96, 1.28) | | 0.976(0.81, 1.18) |
| Middle | | 1.08(0.98, 1.19) | | 1.04(0.87, 1.25) |
| Poorer | | 1.14(0.94, 1.38) | | 0.89(0.74, 1.07) |
| Poorest | | 1 | | 1 |
| **Birth in the last five years** | | | | |
| One | | 1.17(0.97, 1.41) | | 1.02(0.75, 1.37) |
| Two | | 1.20(0.99, 1.45) | | 1.11(0.82, 1.50) |
| More than two | | 1 | | 1 |
| **Frequency of ANC** | | | | |
| ≥4 visits | | 2.65 (2.17,3.23) | | 2.78(2.09, 3.71)** |
| 3 visits | | 2.07(1.68, 2.54) | | 2.18(1.62, 2.92)** |
| 2 visits | | 1.79 (1.43, 2.24) | | 2.10(1.52, 2.91)** |
| 1 visit | | 1 | | 1 |
| **Reading newspaper** | | | | |
| At least once a week | | 1.04(0.82, 1.30) | | 1.19(0.85, 1.68) |
| Less than once a week | | 1.00(0.87, 1.15) | | 1.07(.87, 1.32) |
| Not at all | | 1 | | 1 |
| **Listening to a radio** | | | | |
| At least once a week | | 1.21(1.08, 1.35) | | 1.11(0.94, 1.31) |
| Less than once a week | | 1.06(0.95, 1.19) | | 0.96(0.82, 1.12) |
| Not at all | | 1 | | 1 |
| **Watching television** | | | | |
| At least once a week | | 1.21(1.08, 1.35) | | 0.91(0.73, 1.13) |

*(Continued)*

**Table 6.** (Continued)

| Variable categories | Model I (nul model) | Model II (individual-level factors) | Model III (community-level factors) | Model-IV (full model) |
|---|---|---|---|---|
| | | IRR(95%CI) | IRR(95%CI) | IRR(95%CI) |
| Less than once a week | | 0.88(0.77, 1.01) | | 0.62(0.55, 1.48) |
| Not at all | | 1 | | |
| **Had a mobile phone** | | | | |
| Yes | | 1.01(0.90, 1.13) | | 1.06(0.89, 1.25) |
| No | | 1 | | 1 |
| **Distance to a health facility** | | | | |
| Big problem | | 0.76(0.67, 1.05) | | 0.93(0.89, 1.17) |
| Not a big problem | | 1 | | 1 |
| **Getting permission to go to a health facility** | | | | |
| Not a big problem | | 1.34(1.12, 1.67) | | 1.29(1.03, 1.38)** |
| Big problem | | 1 | | 1 |
| **Covered by health insurance** | | | | |
| No | | 0.85(0.71, 1.02) | | 0.76 (0.68, 0.95)** |
| Yes | | 1 | | |
| **Regions** | | | | |
| Tigray | | | 4.31(3.31, 5.59) ** | 3.84 (2.53, 5.82)** |
| Amhara | | | 3.13(2.39, 4.10) ** | 2.85 (1.86,4.37)** |
| Oromia | | | 2.64(2.01, 3.46) ** | 2.34 (1.87, 3.74)** |
| Somali | | | 1.51(1.12, 2.03) * | 1.96(1.33, 2.88)* |
| Benishangul | | | 2.83(2.13, 3.75) * | 2.54 (1.79, 3.62)* |
| SNNPR | | | 3.14(2.41, 4.10) | 2.76(1.94, 3.93)* |
| Gambella | | | 2.39(1.78, 3.20) | 2.21(1.5, 3.23)* |
| Harari | | | 2.92(2.18, 3.91) | 2.95 (2.08, 4.17)* |
| Addis Ababa | | | 3.51(2.62, 4.70) | 2.82(1.94, 4.09)* |
| Diredawa | | | 2.18(1.61, 2.93) | 2.22(1.54, 3.19)* |
| Afar | | | 1 | |
| **Residence** | | | | |
| Urban | | | 1.21(1.08, 1.37) | 1.13(0.97, 1.43) |
| Rural | | | 1 | 1 |
| **Random effects** | | | | |
| Variance | 0.21 | 0.14 | 0.16 | 0.09 |
| AIC | 14397.74 | 14180.17 | 14222.43 | 13902.28 |
| BIC | 14417.11 | 14419.11 | 14312.84 | 14001.31 |
| MOR | 1.76 | 1.40 | 1.35 | 1.23 |
| PCV | Reference | 0.33 | 0.23 | 0.57 |
| **Model fitness** | | | | |
| Log-likelihood | -7395.86 | -7053.08 | -7097.21 | -6813.13 |
| Deviance | 14791.72 | 14106.16 | 14186.42 | 13626.26.26 |

**Key:** 1: Reference category; IRR = Incidence rate ratio

* Statistically significant at p-value <0.05

receiving BPCR messages as compared to women with a single visit. Similarly, those women with four or more visits had a 2.78 times higher incidence of receiving higher numbers of WHO-recommended BPCR messages than mothers who had only one ANC visit. This could

be because as the number of ANC visits increases, so does the likelihood that women will contact a healthcare provider, which increases the uptake of BPCR messages. This was supported by many studies conducted in Tanzania [20, 42, 43], Nigeria [38, 44], Cameroon [21], Kenya [18], and Ethiopia [45, 46]. This implies that antenatal care visits provide an opportunity to inform pregnant women and assist them in receiving the necessary components of BPCR. As a result, it is critical to educate and train healthcare providers on how to advise pregnant women on BPCR components.

Women who had difficulty in obtaining permission to seek medical care were 37% and 33% less likely to get BPCR messages and a higher number of BPCR messages, respectively. This finding was supported by studies conducted in Ghana [47], and Kenya [48] This could be because when a woman had difficulty obtaining permission to receive medical care, she lost her autonomy, which leads to decreased health-seeking behavior and ease of access to health care providers during pregnancy, all of which resulted in low BPCR message uptake. This implies that institutions that work on women's affairs need to work on empowering women through the enhancement of joint decision-making during seeking medical care.

The likelihood of BPCR message uptake increases as the household wealth index rises, with women in the richest household wealth quintiles nearly twice as likely to receive many BPCR messages as their counterparts in the poorest wealth quantiles. Studies conducted in developing countries like Bangladesh [49], Papua New Guinea [50], Colombia [51], and India [52] documented that maternal health service utilization is higher for women from the richest and richer households compared to those from the poorest households. This could be because women from higher economic classes have more money to visit health facilities [53, 54] and have confidence in prioritizing health service matters during their pregnancy, which could result in a high uptake of BPCR messages. On the other hand, women who belong to the richest household usually have higher educational status [54, 55], access to mass media [56–58], and the ability to spend more money to take frequent ANC visits in which they got adequate BPCR messages. Even though maternal health services in Ethiopia are currently exempt at the facility level, most of the expenses on the way to the health facility, such as transportation and food, are incurred by the clients, making access to maternal health services difficult. As a result, the government needs to work on initiatives to increase women's economic capacity.

The finding showed that employed women had a 33% higher chance of receiving BPCR messages as compared to their counterparts. This was supported by studies conducted in India [31], Nigeria [44], Uganda [59], Rwanda [60], Guinea [61], Angola [62], and Kenya [18]. This could be because unemployed women spend more time at home and are less likely to be exposed to health information about birth preparation than their counterparts. Furthermore, it is well understood that employment status is one factor that greatly influences women's empowerment by increasing autonomy in decision-making in health and the expected incurred costs, all of which increase the uptake of MNCH services [31, 43, 63]. As a result, the government must work to increase employment opportunities for women.

Similarly, those respondents who didn't enroll in a health insurance scheme were 35% less likely to receive BPCR messages as compared to their counterparts. Although a shortage of evidence on the association of this variable with BPCR uptake, some pieces of evidence from studies conducted in India [64], Ghana [65, 66], and Ethiopia [67] showed a positive association with MNCH uptake. This could be because women who are enrolled in health insurance schemes are more likely to visit health facilities and contact healthcare providers. After all, there is no fear of cost, resulting in high demand for maternal health services and a high health-seeking behavior toward MNCH services like BPCR messages. Despite the fact that Ethiopia's maternal health policy exempts MNCH services from out pocket payments, most mothers were not aware of this. Thus, enrolling in health insurance schemes allows them to

become more familiar with existing service delivery points. The finding lends support to the notion that removing direct financial barriers through health insurance schemes may increase maternal healthcare utilization in Ethiopia [66, 68, 69]. In addition to maternal care issues, health insurance may act as a pro-poor equalizing agent in health financing by increasing access to health care [69]. Hence, the local administrative authorities should make efforts to make households enroll in health insurance schemes.

The findings from the current study were based on the analysis of nationally representative data from a large sample size collected with standardized and validated data collection instruments and methodology, making the findings more generalizable. In addition, due to the clustering effect of EDHS data, a multilevel-modeling technique was used in the analysis, which provides disaggregated evidence on individual and community-level determinants for designing contextual interventions. Furthermore, the analysis took into account both overall BPCR uptake (as a categorical variable) and WHO-recommended BPCR message items as count data, and thus the findings could be used as input at the local and policy levels to strengthen BPCR message delivery and quality.

Even with the aforementioned advantages, the study is not beyond limitations. First off, because the data were obtained through retrospective interviews of a selected group of women who had a live birth within five years of the survey, the findings may be prone to recall bias. Second, because the data is cross-sectional, the findings are likely to suffer from social desirability bias and fail to address the cause-effect relationship ship. Even though our findings are consistent with the existing evidence, there could be other factors mediating or confounding the current associations.

## Conclusion

The level of BPCR message uptake in Ethiopia was found to be low. In addition, compliance with the WHO-recommended elements of PBCR messages is too low. The frequency of ANC visits and the ease of obtaining permission to seek medical care were significantly associated with both the overall magnitude and the mean number of BPCR message uptake. Women's employment status, household wealth index, and enrollment in health insurance schemes, on the other hand, were found to be significantly associated with a good uptake of BPCR message uptake. Managers in the health sector and healthcare providers must work to increase the number of ANC visits. Policymakers should prioritize the implementation of activities and interventions that increase women's empowerment, particularly their autonomy in decision-making, job opportunity, and economic capability to enhance their health-seeking behavior during pregnancy. Finally, the local administrative bodies should make efforts to make households enroll in health insurance schemes.

## Acknowledgments

We are grateful to ICF macro (Calverton, USA) for providing the 2016 DHS data of Ethiopia.

## Author Contributions

**Conceptualization:** Aklilu Habte.

**Data curation:** Aklilu Habte, Aiggan Tamene.

**Formal analysis:** Aklilu Habte, Aiggan Tamene, Demelash Woldeyohannes.

**Methodology:** Aklilu Habte, Aiggan Tamene, Demelash Woldeyohannes.

**Resources:** Aklilu Habte.

**Software:** Aklilu Habte.

**Visualization:** Aklilu Habte.

**Writing – original draft:** Aklilu Habte, Demelash Woldeyohannes.

**Writing – review & editing:** Aklilu Habte, Aiggan Tamene, Demelash Woldeyohannes.

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
