## [Decision Letter · Decision Letter 0]

25 Nov 2022

PONE-D-22-24552The uptake of WHO-recommended Birth Preparedness and complication readiness messages and its determinants in Ethiopia: A multilevel mixed-effects analyses of the 2016 Ethiopian demographic health surveyPLOS ONE

Dear Dr. Habte Hailegebireal,

Thank you for submitting your manuscript to PLOS ONE. After careful consideration, we feel that it has merit but does not fully meet PLOS ONE’s publication criteria as it currently stands. Therefore, we invite you to submit a revised version of the manuscript that addresses the points raised during the review process.

We look forward to receiving your revised manuscript.

Kind regards,

Melkamu Merid Mengesha, MPH

Academic Editor

PLOS ONE

Journal Requirements:

Reviewers' comments:

Reviewer's Responses to Questions

**Comments to the Author**

1. Is the manuscript technically sound, and do the data support the conclusions?

Reviewer #1: Partly

Reviewer #2: Yes

2. Has the statistical analysis been performed appropriately and rigorously? 

Reviewer #1: Yes

Reviewer #2: Yes

3. Have the authors made all data underlying the findings in their manuscript fully available?

Reviewer #1: No

Reviewer #2: Yes

4. Is the manuscript presented in an intelligible fashion and written in standard English?

Reviewer #1: Yes

Reviewer #2: Yes

5. Review Comments to the Author

Reviewer #1: General comments

1. Included samples (women) are not specified whether they are ever pregnant within 5 years preceding a survey or pregnant during the survey

2. In the result (both in the abstract and main result) the result of the random effect is not described.

Methods

1. Major issue: The authors used t-test and ANOVA to associate the frequency of BPCR, however, they suggest a mixed effect model as best fitted data for their data. So, how does data that didn’t violate the assumption for t-test (independence of observation) and ANOVA (independence of observation, equal variance) do violet for multivariate? Did you consider autocorrelation?

2. Authors mentioned “Those variables with p-values less than 0.05 were eligible for a multilevel mixed-effect negative binomial regression using a generalized linear model (GLM) to determine the determinants of the number of BPCR messages received”. If they already exclude non-significant variables there is selection bias as they consider significant variables only. At least there should be inclusive criteria to include some variables with low p-value (significant and non-significant) using the rule of thumb or any selection criteria.

3. Under the model selection part: there first six lines are not about model selection. It is all about the random effect model and should be put under the statistical analysis part.

4. Model selection: the overdispersion might be by chance and the authors should indicate whether the variance is actually higher than the mean (using the rule of thumb or hypothesis testing)

5. Model selection is choosing the parsimonious model from possible fitted models. hence before choosing the best-fitted, it is better if you check for assumption (zero inflation nature of variables not by likelihood but nature of data (indicating whether they include women who only had ANC follow-up))

Result

1. It is strange that the outcome is described before the basic characteristics of the participants. describe sociodemographic characteristics before “The Overall uptake of WHO-recommended BPCR messages”

2. It is also recommendable that authors used a flow chart to show % of observations included in the final analysis (sample size >>excluded (reason)>>included into descriptive analysis>>included into outcome one (uptake)>>included for frequency of uptake etc.)

3. Table one might be described in the graphs and narration of some results

4. For table 2, what type of estimation was used? describe how this estimation is done in the methodology. Is this estimation for all Ethiopian mothers? if not why didn't simply describe the summary statistics for the sample? Contraceptive utilization: what if contraceptive is used after pregnancy? Describe how contraceptive is measured and used in your data.

5. Mixed model result: the base for most of the variables (region, religion, …) is not described in the method. So where is the variable selection described in the method part?

6. In table 5: the result of religion showed that protestant is increasing the odds of uptalking BPCR. This result is neither described in the result nor in the discussion. My question is what is the explanation for this result? did you check the adequacy of power and sample size to include all these variables?

7. Table 5: what is the denominator to calculate %? aren't we expecting total and respective freq. for each category?

8. Enrollment in health insurance schemes decrease the odds of BPCR message uptake by 45%, so what will be the implication of this finding? What is the recommendation (will you recommend against the enrollment or any explanation for this outcome?)

Discussion

1. Most of the reason you described is weak. Justify the discrepancies or associations in terms of scientific reasoning and theories do not guess (e.g., you justified the discrepancy of prevalence as the difference in sociodemographic characteristics of study participants, and health-system-related characteristics (like shortage of health care providers and distance to health facilities: aren't you included those factors in the model (already controlled)?

AND

Furthermore, the difference could also be due to differences in study population coverage,

study setting, and time (didn’t you weighted the data? Then why didn’t you choose the study conducted at the same time with your data?)

2. Limitation: did you include all the possible factors that could predict your outcome? If not, why don't you mention under limitation?

Reviewer #2: Comment 1: Table1: Number of BPCR messages received by women during their recent pregnancy

preceding the survey in Ethiopia, 2016.

1. Say something about the table since audiences/Readers did not understand the meaning.

Comment 2: Do not use abbreviation in the abstract part of the document.

Comment 3:Describe EDHS 2016 Collected data under Ethical considerations.

Comment 4:The study relied on population-based, nationally representative data from 2016 Ethiopian

Demographic and Health Survey (DHS), the seventh in a series of national-level population

and health surveys carried out as part of the global Demographic and Health Survey (DHS)

program. So, why only use EHDS 2016 Data for this study since the title on WHO Recommendations on BPCR messages.

Comment 5: this paper did not describe about the population characteristics.

Comment 6: Is there any exclusion data? not described

Comment 7: Did you check Completeness of Data?

Comment 8: under limitation of the study, it says ''retrospective interviews of a selected group of women who had a live birth

within five years of the survey''. Rewrite it.

Comment 9: What were/was drawback of EDHS2016 So far? you did not well explained.

Comment 10: Discussion should be rewrite it again.

6. PLOS authors have the option to publish the peer review history of their article (what does this mean?). If published, this will include your full peer review and any attached files.

Reviewer #1: No

Reviewer #2: No

---

## [Author Response · Author response to Decision Letter 0]

27 Nov 2022

All comments and questions raised by the editor and reviewers were addressed and submitted as a file "Response to reviewers" in the submission system.

---

## [Decision Letter · Decision Letter 1]

19 Jan 2023

PONE-D-22-24552R1The uptake of WHO-recommended birth preparedness and complication readiness messages during pregnancy and its determinants among Ethiopian women: A multilevel mixed-effect analyses of 2016 demographic health surveyPLOS ONE

Dear Dr. Habte Hailegebireal,

Thank you for submitting your manuscript to PLOS ONE. After careful consideration, we feel that it has merit but does not fully meet PLOS ONE’s publication criteria as it currently stands. Therefore, we invite you to submit a revised version of the manuscript that addresses the points raised during the review process.

We look forward to receiving your revised manuscript.

Kind regards,

Melkamu Merid Mengesha, MPH

Academic Editor

PLOS ONE

Journal Requirements:

Additional Editor Comments:

line 2-3: remove commas between texts in the title.

lines 34-36: This statement starting in lines 34-36 in the abstract is not supported by evidence as the government of Ethiopia in partnership with international actors did a huge investment to improve maternal health.

Please relook this statement and provide it in appropriate terms.

line 50: expand abbreviations (CI and others in the document) on first use. use 'colon (:) instead of the 'semi-colon (;)'' to list the upper and lower confidence limits as in like [95% CI: X, Y].

Interpretation of results: present interpretations for individual and group level factors separately; do the same in the conclusion too. Interpretation of results should carefully reflect the nature of design and model used.

Reviewers' comments:

Reviewer's Responses to Questions

**Comments to the Author**

1. If the authors have adequately addressed your comments raised in a previous round of review and you feel that this manuscript is now acceptable for publication, you may indicate that here to bypass the “Comments to the Author” section, enter your conflict of interest statement in the “Confidential to Editor” section, and submit your "Accept" recommendation.

Reviewer #1: All comments have been addressed

Reviewer #2: All comments have been addressed

2. Is the manuscript technically sound, and do the data support the conclusions?

Reviewer #1: Yes

Reviewer #2: Yes

3. Has the statistical analysis been performed appropriately and rigorously? 

Reviewer #1: Yes

Reviewer #2: Yes

4. Have the authors made all data underlying the findings in their manuscript fully available?

Reviewer #1: Yes

Reviewer #2: Yes

5. Is the manuscript presented in an intelligible fashion and written in standard English?

Reviewer #1: Yes

Reviewer #2: Yes

6. Review Comments to the Author

Reviewer #1: (No Response)

Reviewer #2: (No Response)

7. PLOS authors have the option to publish the peer review history of their article (what does this mean?). If published, this will include your full peer review and any attached files.

Reviewer #1: **Yes: **Lemma Demissie

Reviewer #2: **Yes: **Feleke Gebremeskel W/Hawariat

---

## [Author Response · Author response to Decision Letter 1]

19 Jan 2023

Comment 1: line 2-3: remove commas between texts in the title.

Response: thank you for your meticulous review, and we have corrected the statement by replacing the comma with the colon in the "title page" section of the “Revised manuscript with track changes”, Line 2, Page 1.

Comment 2: Lines 34-36: This statement starting in lines 34-36 in the abstract is not supported by evidence as the government of Ethiopia in partnership with international actors did a huge investment to improve maternal health.

Response: Thank you for your comment and suggestion, and we have corrected the statement and highlighted it in the "Abstract" section of the “Revised manuscript with track changes”, Line 34-35, Page 2.

Comment 3: expand abbreviations (CI and others in the document) on first use. use 'colon (:) instead of the 'semi-colon (;)'' to list the upper and lower confidence limits as in like [95% CI: X, Y].

Response: We appreciate your meticulous review and we already have written the expanded version for the abbreviation in the "Abstract" section of the “Revised manuscript with track changes”, Line 48, Page 2. The punctuation issue also has been corrected and highlighted in Line 50, page 2

Comment 4: Interpretation of results: present interpretations for individual and group level factors separately; do the same in the conclusion too. Interpretation of results should carefully reflect the nature of the design and model used.

Response: thank you for your vital comment that helps us to make the results of multi-level analyses clear to the readers by showing the nature of the design and model used. Accordingly, we added the individual and community-level factors that were significantly associated with the uptake of BPCR messages in the "Abstract" section of the “Revised manuscript with track changes”, Line 52-60, Page 2. 

END________________________________________

 THANK YOU!!!

---

## [Editor Report · Decision Letter 2]

5 Feb 2023

PONE-D-22-24552R2The uptake of WHO-recommended birth preparedness and complication readiness messages during pregnancy and its determinants among Ethiopian women: A multilevel mixed-effect analyses of 2016 demographic health surveyPLOS ONE

Dear Dr. Habte Hailegebireal,

Thank you for submitting your manuscript to PLOS ONE. After careful consideration, we feel that it has merit but does not fully meet PLOS ONE’s publication criteria as it currently stands. Therefore, we invite you to submit a revised version of the manuscript that addresses the points raised during the review process.

We look forward to receiving your revised manuscript.

Kind regards,

Melkamu Merid Mengesha, MPH

Academic Editor

PLOS ONE

Journal Requirements:

Additional Editor Comments:

The length of the abstract is too much beyond the recommended 300 words count in the revised submission. Please check author PLOS ONE author guideline and revise this as a final opportunity to look into the abstract and other unnoticed issues (including editorials) throughout the document.

<quillbot-extension-portal></quillbot-extension-portal>

---

## [Author Response · Author response to Decision Letter 2]

5 Feb 2023

Thank you for your thorough review; we have corrected the abstract section per the journal's requirements. In your previous comment, you suggested that individual and community level factors be reported in the abstract, and this was the sole reason that the abstract section was lengthened. Overall, we have attempted to reduce the lengthy abstract accordingly.

---

## [Editor Report · Decision Letter 3]

23 Feb 2023

The uptake of WHO-recommended birth preparedness and complication readiness messages during pregnancy and its determinants among Ethiopian women: A multilevel mixed-effect analyses of 2016 demographic health survey

PONE-D-22-24552R3

Dear Dr. Habte Hailegebireal,

We’re pleased to inform you that your manuscript has been judged scientifically suitable for publication and will be formally accepted for publication once it meets all outstanding technical requirements.

Kind regards,

Melkamu Merid Mengesha, MPH

Academic Editor

PLOS ONE

Additional Editor Comments (optional):

Reviewers' comments:

<quillbot-extension-portal></quillbot-extension-portal>

---

## [Editor Report · Acceptance letter]

14 Mar 2023

PONE-D-22-24552R3 

The uptake of WHO-recommended birth preparedness and complication readiness messages during pregnancy and its determinants among Ethiopian women: A multilevel mixed-effect analyses of 2016 demographic health survey 

Dear Dr. Habte Hailegebireal:

I'm pleased to inform you that your manuscript has been deemed suitable for publication in PLOS ONE. Congratulations! Your manuscript is now with our production department. 

Kind regards, 

on behalf of

Mr. Melkamu Merid Mengesha 

Academic Editor

PLOS ONE